# Comparison of neoadjuvant treatment and surgery first for resectable or borderline resectable pancreatic carcinoma: A systematic review and network meta-analysis of randomized controlled trials

**Lu Huan[1], Fucai Yu[1], Ding Cao[1], Hantao Zhou[1], Maoling Qin[2], Yang Cao**[1]*

**1** Department of Hepatopancreatobiliary Surgery, Chongqing Fifth People's Hospital, Chongqing, Chongqing, China, **2** Department of Hepatobiliary Surgery, The Second Affiliated Hospital of Chongqing Medical University, Chongqing, Chongqing, China

* teacher_caoyang@163.com

## Abstract

### Background

Current treatment recommendations for resectable or borderline pancreatic carcinoma support upfront surgery and adjuvant therapy. However, neoadjuvant therapy (NT) seems to increase prognosis of pancreatic carcinoma and come to everyone's attention gradually. Randomized controlled trials offering comparison with the NT are lacking and optimal neoadjuvant treatment regimen still remains uncertain. This study aims to compare both treatment strategies for resectable or borderline resectable pancreatic cancer.

### Methods

The PRISMA checklist was used as a guide to systematically review relevant peer-reviewed literature reporting primary data analysis. We searched PubMed, Medline, EMBASE, Cochrane Datebase and related reviews for randomized controlled trials comparing neoadjuvant therapy with surgery first for resectable or borderline resectable pancreatic carcinoma. We estimated relative hazard ratios (HRs) for median overall survival and ratios risks (RRs) for microscopically complete (R0) resection among different neoadjuvant regimens and major complications. We assessed the effects of neoadjuvant therapy on R0 resection rate and median overall survival with Bayesian analysis.

### Results

Thirteen eligible articles were included. Eight studies performed comparison neoadjuvant therapy with surgery first, and R0 resection rate was recorded in seven studies. Compared with surgery first, neoadjuvant therapy did increase the R0 resection rate (RR = 1.53, $I^2$ = 0%, P< 0.00001), there was a certain possibility that gemcitabine + cisplatin (Gem+Cis) + Radiotherapy was the most favorable in terms of the fact that there was no significant

**Funding:** The author(s) received no specific funding for this work.

**Competing interests:** The authors have declared that no competing interests exist

difference concerning the results from the individual studies. In direct comparison, four studies were included and estimated that Neoadjuvant therapy improved mOS compared with upfront surgery (HR 0.68, 95% CI 0.58–0.92; P = 0.012; $I^2$ = 15%), after Bayesian analysis it seemed that regimen with Cisplatin/ Epirubicin then Gemcitabine/ Capecitabine (PEXG) was most likely the best with a relatively small sample size. The rate of major surgical complications was available for six studies and ranged from 11% to 56% with neoadjuvant therapy and 11% to 45% with surgery first. There was no significant difference between neoadjuvant therapy and surgery first, also with a high heterogeneity (RR = 0.96, 95%CI = 0.65–1.43; P = 0.85; $I^2$ = 46%).

## Conclusion

In conclusion neoadjuvant therapy might offer benefit over up-front surgery. Neoadjuvant therapy increased the R0 resection rate with gemcitabine + cisplatin + Radiotherapy that was the most favorable and improved mOS with Cisplatin/ Epirubicin then Gemcitabine/ Capecitabine (PEXG) that was most likely the best.

## Introduction

Pancreatic cancer is the fourth leading cause of cancer-related mortality in the United States [1, 2]. Although pancreatic cancer can be treated with curative surgery by complete resection (R0), the 5-year survival rate remains low at 15%-20% [3, 4]. Moreover, the incidence of pancreatic cancer is on the rise globally, and it is projected to become the second leading cause of cancer-related deaths among all malignancies in Western countries by 2030.

Although preoperative (neoadjuvant) chemotherapy was a standard method of treatment for many other types of cancer, its benefits in treating resectable or borderline resectable pancreatic cancer patients had not yet been established [5, 6]. Meanwhile, for the current treatment situation, adjuvant chemotherapy after resection was the gold standard of treatment for resectable pancreatic cancer [3, 4]. The National Comprehensive Cancer Network guidelines recommended neoadjuvant therapy for borderline resectable cancer, while NICE guidelines only recommended neoadjuvant therapy as part of a clinical trial [3, 4]. The recommendations in both guidelines were not based on randomized controlled trials (RCTs).

A recent study [7] based on Surveillance, Epidemiology, and End Results (SEER) data from nearly 4000 patients showed that neoadjuvant radiotherapy (with or without chemotherapy) had better survival benefits compared to direct surgical treatment (with or without adjuvant therapy). Additionally, a recent meta-analysis of randomized controlled trials (RCTs) demonstrated that neoadjuvant therapy improved overall survival (OS) in patients with borderline resectable pancreatic cancer compared to up-front surgical treatment (US), but with significant heterogeneity [8]. However, comparing OS between neoadjuvant therapy and direct surgery in research studies is challenging and complex [9]. Furthermore, most published meta-analyses [8, 10, 11] reported higher R0 resection rates overall with neoadjuvant therapy, contradicting the clear benefit of neoadjuvant therapy. Despite recommendations based on several centers, including retrospective studies and small phase II trials, suggesting neoadjuvant therapy for patients with borderline resectable pancreatic cancer, large-scale randomized controlled trials or studies focusing on postoperative complications have not been included, and the specific type of neoadjuvant treatment has not been mentioned or analyzed [8, 12–14].

The primary aim of our current meta-analysis was to evaluate various neoadjuvant treatment regimens and the "surgery first" approach for patients with resectable or borderline resectable pancreatic cancer. We specifically focused on randomized controlled trials (RCTs) and aimed to identify the most effective neoadjuvant treatment regimen available.

## Methods

Ethical approval or patient consent was not required since the present study was a review of previously published literature. And there is no conflicts of interests.

### Search strategy and selection criteria

In accordance with the PRISMA guidelines (Preferred Reporting Items for Systematic Reviews and Meta-Analysis) [15], a comprehensive systematic review was conducted to analyze the existing literature on neoadjuvant treatment for resectable pancreatic carcinoma or borderline resectable pancreatic carcinoma (BRPC) followed by surgery. The search was conducted on PubMed, Embase, and Web of Science, without any language or publication date restrictions. The search strategy involved the use of various keywords and Medical Subject Heading (MeSH and Emtree) terms, such as "pancreatic carcinoma," "pancreatic cancer," and "neoadjuvant treatment." These terms were combined using the operators "AND" or "OR" to refine the search results. (Fig 1, Flow diagram showing the selection of randomized controlled trials).

In order to conduct the comprehensive meta-analysis, it was essential to include a variety of studies that explore different neoadjuvant therapy regimens followed by surgery or compare neoadjuvant treatment with up-front surgery. Additionally, it was important to ensure that the studies selected provide the full text rather than just abstracts or conference presentations. To maintain the integrity of the analysis, we had excluded studies that compare the same group of patients before and after pancreatic surgery. Furthermore, studies focusing on locally advanced (LA) pancreatic carcinoma had also been excluded. Other criteria for exclusion include retrospective and non-randomized prospective studies, as well as those that were not written in English or were only published as conference abstracts.

### Data extraction and methodological quality

The information provided below was obtained from search results and served as a basis for determining the possibility of comparability. To assess comparability, various criteria were considered, including author name, publication year, journal name, title, inclusion and exclusion criteria, primary and secondary endpoints. In this study, two authors, Hantao Zhou and Ding Cao, independently extracted and verified the data. They meticulously examined the number of patients, neoadjuvant treatment regimen, resectable rate (R0), median overall survival (mOS) in months, Hazard Ratio (HR), and major complications. Additionally, when analyzing studies from the same trial but with different follow-up periods, only the data from the most recent study were utilized. To evaluate the risk of bias, all studies were assessed using a standardized list of seven potential risks, based on the Cochrane Collaboration's tool for assessing bias. During this assessment, various factors were considered, and each study was categorized as having low, high, or unclear risk of bias [16] (Fig 2, Risk bias of graph. Each risk of bias item presented as percentages across all of the included trials, which indicated the proportion of different level risk of bias for each item. Fig 3, Risk bias of summary. Judgments about each risk of bias item for each included trials. Green indicates low risk of bias. Yellow indicates unclear risk of bias. Red indicates high risk of bias).

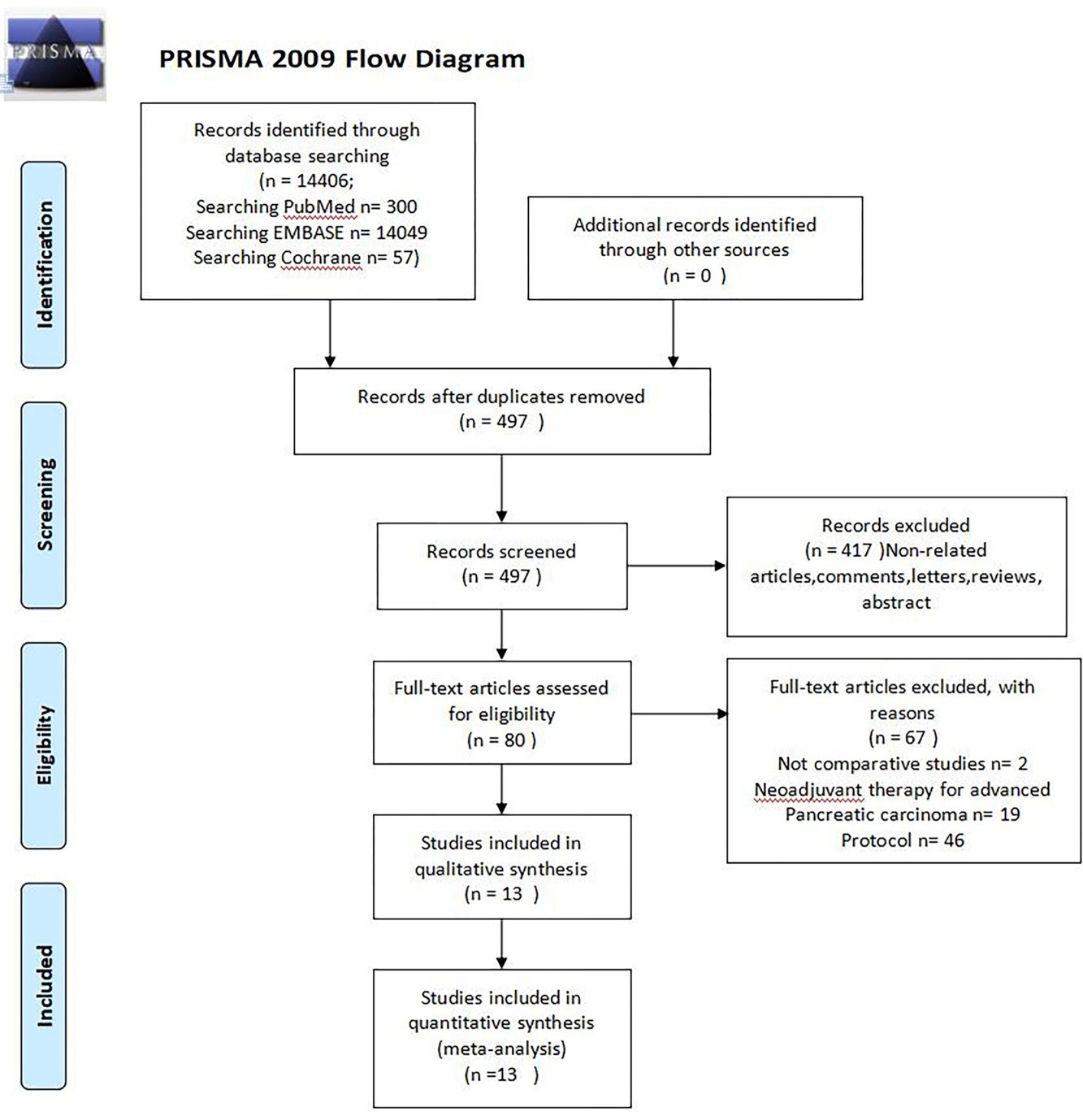

**Fig 1. Flow diagram.** The flow diagram showed the selection of randomized controlled trials.

## Data synthesis and analysis

Our main focus was on achieving R0 resection, while also considering median overall survival and major complications as secondary outcomes. To compare the effectiveness of different neoadjuvant therapy regimens, even those without direct comparisons, we conducted a Bayesian network meta-analysis using WinBUGS 1.4 (MRC Biostatistics Unit), Revman 5.4, and

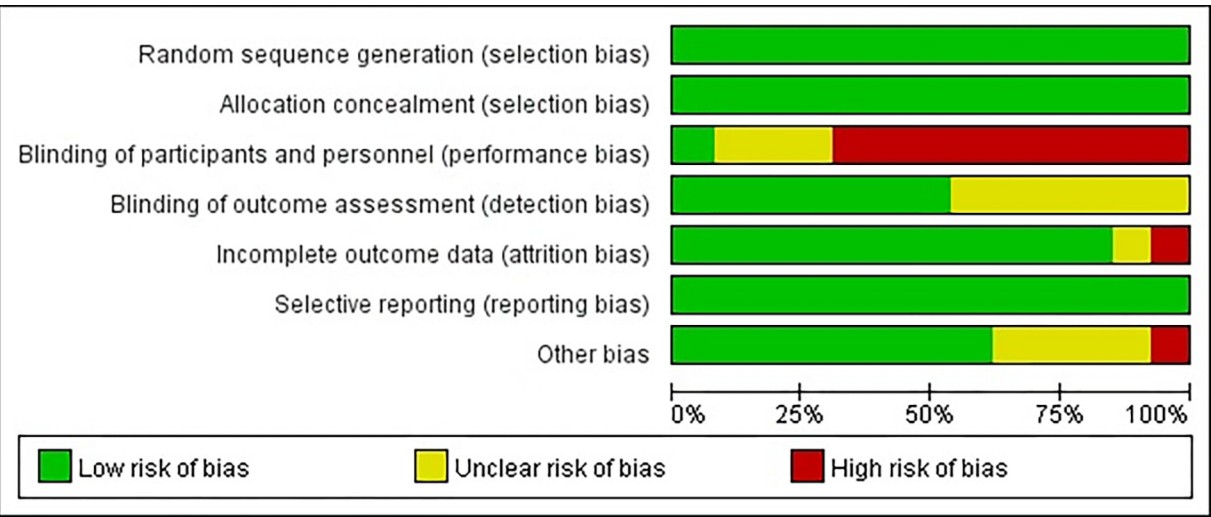

**Fig 2. Risk bias of graph.** Each risk of bias item presented as percentages across all of the included trials, which indicated the proportion of different level risk of bias for each item.

STATA 14. In WinBUGS, we utilized non-informative uniform and normal prior distributions, along with three sets of starting values, resulting in 150,000 iterations (50,000 per chain) to obtain posterior distributions of model parameters. For Revman 5.4, we employed the Ratio Risk (RR) method to estimate dichotomous variables reported in the original studies. To account for heterogeneity among the studies, we employed random-effects models for the meta-analysis. When analyzing median overall survival, we preferred adjusted hazard ratios (HRs) as our outcome measure, as HRs consider censoring, provide time-to-event information, and account for adjusted confounders. In cases where HRs were not reported, we estimated them using the method described by Tierney and colleagues [17]. To assess the consistency between direct and indirect comparisons, it was crucial to evaluate the conflict between direct and indirect evidence in our network meta-analysis [18]. In line with the NICE decision-support documents, we measured inconsistency by comparing deviance residuals and deviance information criteria (DIC) statistics in fixed consistency and inconsistency models [18, 19]. We plotted the posterior mean deviance of individual data points in the inconsistency model against their posterior mean deviance in the consistency model to identify any loops in the treatment network indicating inconsistency [19]. To calculate pooled effect estimates in the meta-analysis, we constructed random-effects models due to suspicions of statistical or clinical heterogeneity. We categorized the $I^2$ levels as low when less than or equal to 25%, moderate when ranging between 25% and 50%, and high when greater than 50% [20]. For all analyses, statistical significance was set at $p < 0.05$.

Our meta-analysis focused on the comparison between neoadjuvant therapy which includes neoadjuvant chemotherapy and/or radiation therapy and up-front surgery for pancreatic carcinoma. Additionally, we also examined the effectiveness of different neoadjuvant regimens in treating pancreatic carcinoma.

## Results

Out of the 497 articles that were identified, a total of 417 were chosen for a thorough review of their full texts. Among these, 13 randomized controlled trials (RCTs) [13, 21–32] were included in the analysis. These trials focused on comparing up-front surgery (US) with

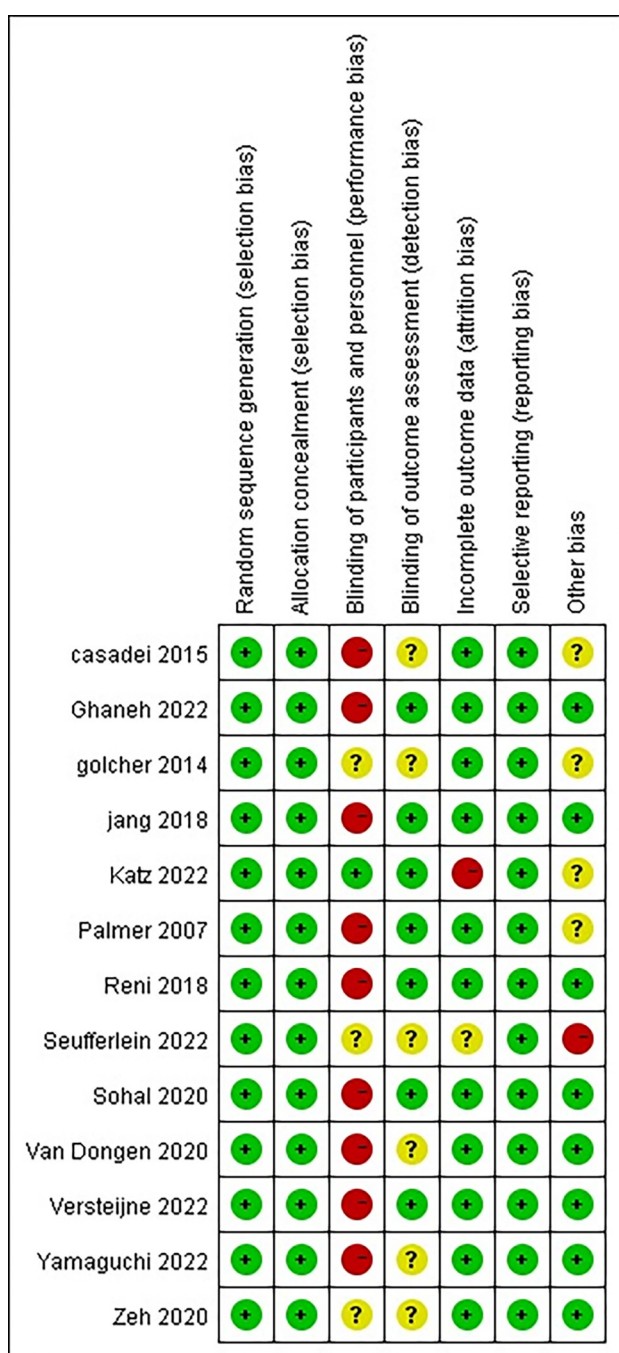

**Fig 3. Risk bias of summary.** Judgments about each risk of bias item for each included trials. Green indicates low risk of bias. Yellow indicates unclear risk of bias. Red indicates high risk of bias.

neoadjuvant therapy for pancreatic carcinoma (PDAC), as well as comparing different regimens of neoadjuvant therapy. Table 1 provides a detailed overview of the characteristics of the trials that were included in the analysis. In total, there were 1159 patients who were randomized to either receive neoadjuvant therapy (NT) or undergo up-front surgery (US). The majority of patients, 743 (64%), were assigned to the NT group, while 416 (36%) were assigned to the US group. In terms of the specific neoadjuvant therapy regimens used in the trials, six of them

**Table 1. Study characteristics.**

| Study (Year) | Country | Comparison | No | Intervention(cycles) | Criteria arterial | Resectability status |
|---|---|---|---|---|---|---|
| Casadei (2015) | Italy | NACR vs US | 18 | Neoadj. gemcitabinebased CRT (54 Gy) (12 months)+ US+ adj. gemcitabine (6) | No contact with HA/ CA SMA | RPC |
| | | | 20 | US+ Adj. gemcitabine(6) | | |
| Ghanem (2022) | UK and Germany | NAC vs NAC vs NACR vs US | 20 | Neoadj gemcitabine/ capecitabine (2)+ US+ adj. gemcitabine or adj. 5-FU/FA (6) | 2013 NCCN: HA encasement allowed, tumor abutment with SMA≤ 180˚ | BRPC |
| | | | 20 | Neoadj. mFOLFIRINOX(4)+ US+ adj. gemcitabine for adj. 5-FU/FA (6) | | |
| | | | 16 | Neoadj. capecitabine-based CRT (50.4Gy)+ US + adj. gemcitabine or adj. 5-FU/ FA (6) | | |
| | | | 32 | US+adj. gemcitabine or adj. 5-FU/FA (6) | | |
| Golcher (2015) | Germany, Switzerland and Austria | NACR vs US | 33 | Neoadj. gemcitabine/ cisplatin-based CRT (3D-conformal plans, 55.8Gy) (6 months)+ US+ adj. gemcitabine (6) | HA /SMA/ CA≤180˚ | BRPC |
| | | | 33 | US+ Adj. gemcitabine (6) | | |
| Jang (2018) | South Korea | NACR vs US | 27 | Neoadj. gemcitabine-based CRT (3D-conformal plans, 54 Gy) (6 months)+ US+ adj. gemcitabine (4) | 2012 NCCN: HA encasement allowed, tumor abutment with SMA ≤ 180˚ | BRPC |
| | | | 23 | US+ adj. gemcitabine-based (4) CRT (3D-conformal plans, 54 Gy) | | |
| Katz (2022) | Canada and USA | NAC vs NACR | 70 | Neoadj. mFOLFIRINOX(8)+ US+ adj. FOLFOX (4) | HA reconstructible, tumor abutment with SMA < 180˚ | BRPC |
| | | | 56 | Neoadj. mFOLFIRINOX-based CRT (hypofractionated)(7)+ US+adj. mFOLFOX (4) | | |
| Palmer (2007) | UK | Gem vs Gem + Cis | 24 | Neoadj. Gemcitabine (6) alone | Absence of invasion in HA/ CA/ SMA | RPC |
| | | | 26 | Neoadj. gemcitabine/ cisplatin (6) | | |
| Reni (2018) | Italy | NAC vs US | 32 | Neoadj. gemcitabine/cisplatin/ epirubicin/ capecitabine (3)+ US+ adj. gemcitabine/ cisplatin/ epirubicin/ capecitabine (3) | Absence of invasion in HA/ CA/ SMA | RPC |
| | | | 30 | US+ adj. gemcitabine/ cisplatin/ epirubicin/ capecitabine (6) | | |
| Seufferlein (2023) | Germany | NAC vs US | 59 | Neoadj. gemcitabine nab- Paclitaxel (2)+ US+ adj. gemcitabine nab- Paclitaxel(4) | clear fat planes around the celiac artery, hepatic artery and superior mesenteric artery | RPC |
| | | | 59 | US+ adj. gemcitabine nab- Paclitaxel(6) | | |
| Sohal (2021) | USA | NAC+ US vs NAC+ US | 55 | Neoadj. mFOLFIRINOX (6)+ US+ adj. mFOLFIRINOX (6) | No interface of HA/CA/SMA | RPC |
| | | | 47 | Neoadj. gemcitabine/nabpaclitaxel (9)+ US+ adj. gemcitabine/nab-paclitaxel (9) | | |
| van Dongen (2020) | Netherlands | NAC vs US | 66 | Neoadj. Gemcitabine (3)+ radiotherapy (15 fractions of 2.4 Gy) US+ adj. gemcitabine(4) | No contact with HA/ CA/ SMA | RPC/ BRPC |
| | | | 98 | US+ adj. gemcitabine/nabpaclitaxel (6) | | |
| Versteijne (2022) | Netherlands | NACR vs US | 119 | Neoadj. gemcitabine/nabpaclitaxel (2)+ US+adj. gemcitabine/nab-paclitaxel (4) | RPC: no arterial contact BRPC: arterial contact≤90˚ | RPC/ BRPC |
| | | | 127 | US+ adj. gemcitabine/nabpaclitaxel (6) | | |
| Yamaguchi (2022) | Japan | NAC+ US vs NAC+ US | 26 | Neoadj. mFOLFIRINOX (4)+ US+ adj. S-1 (6 months) | 2015 NCCN: CA/ SMA180˚ (body/ tail) | BRPC |
| | | | 25 | Neoadj. gemcitabine/nabpaclitaxel (2)+ US+ adj. S-1 (6 months) | | |
| Zeh (2020) | USA | NAC+ US vs NAC+ US | 34 | Neoadj. gemcitabine/nabpaclitaxel (2) + hydroxychloroquine + US | 2015 NCCN: CA/ SMA180˚ (body/ tail) | RPC/ BRPC |
| | | | 30 | Neoadj. gemcitabine/nabpaclitaxel (2) + US | | |

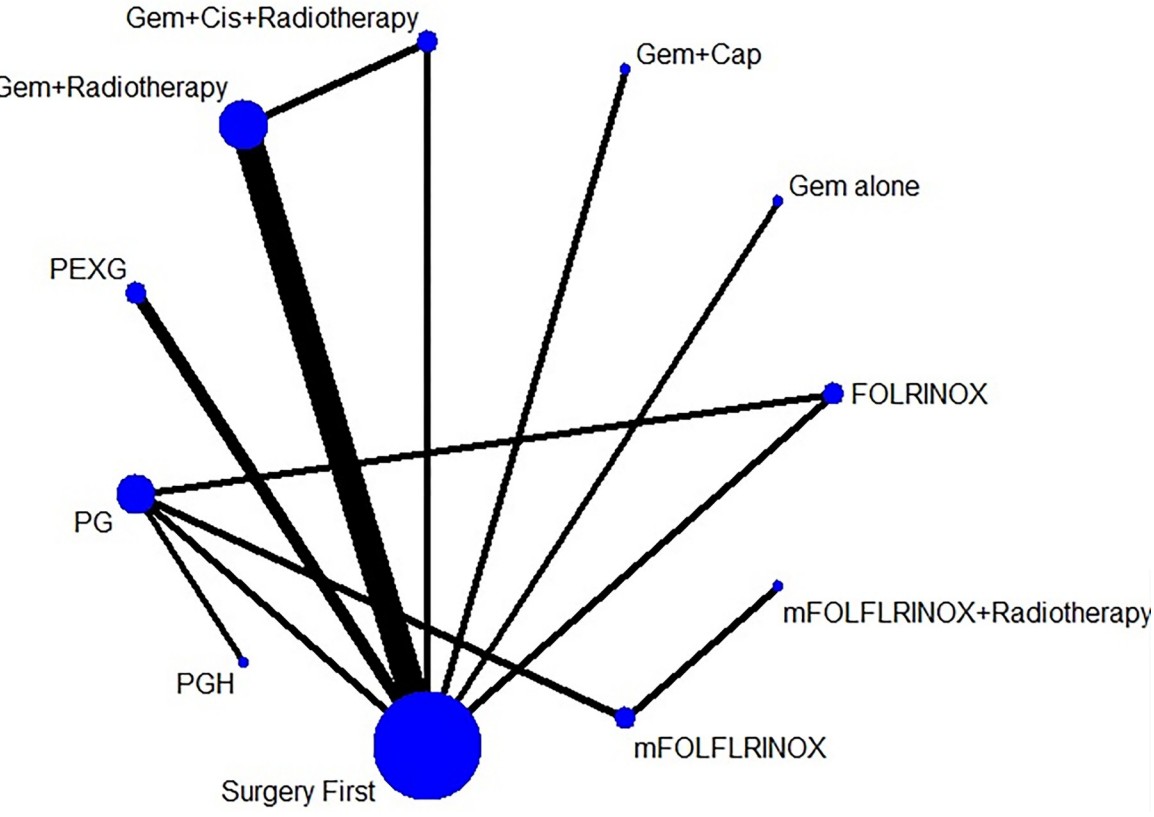

**Fig 4. Network of the comparisons for the Bayesian network meta-analysis.** The size of the nodes is proportional to the number of patients (in parentheses) randomised to receive the treatment. The width of the lines is proportional to the number of trials (beside the line) comparing the connected treatments. (Gem: Gemcitabine; Cis: Cisplatin; PEXG: Gemcitabine + Cisplatin then oral capecitabine; mFOLFLRINOX: Ooxaliplatin + irinotecan + 5-fluorouracily; PG: Gemcitabine + nab-paclitaxel; PGH: Gemcitabine + nab-paclitaxel + hydroxychloroquine; FOLFLRINOX: Oxaliplatin + leucovorin + irinotecan.

utilized neoadjuvant chemoradiotherapy (CRT), while eight trials focused on systemic chemotherapy. The NT regimens included gemcitabine-based therapy, Ooxaliplatin+ irinotecan + 5-fluorouracily (mFOLFIRINOX), fluorouracil, leucovorin, irinotecan, and oxaliplatin (FOLFIRINOX), Cisplatin/ Epirubicin then Gemcitabine/ Capecitabine (PEXG), gemcitabine/ nab-paclitaxel (PG), and nab-paclitaxel/ gemcitabine/ hydroxychloroquine (PGH) (Fig 4). Network of the comparisons for the Bayesian network meta-analysis. The size of the nodes is proportional to the number of patients (in parentheses) randomised to receive the treatment. The width of the lines is proportional to the number of trials (beside the line) comparing the connected treatments. (Gem: Gemcitabine; Cis: Cisplatin; PEXG: Gemcitabine + Cisplatin then oral capecitabine; mFOLFLRINOX: Ooxaliplatin + irinotecan + 5-fluorouracily; PG: Gemcitabine + nab-paclitaxel; PGH: Gemcitabine + nab-paclitaxel + hydroxychloroquine; FOLFLRINOX: Oxaliplatin + leucovorin + irinotecan).

## R0 resection

The primary outcome of interest in this study was the R0 resection rate, which was available for twelve studies. Among these, eight studies compared neoadjuvant therapy followed by surgery with surgery alone. However, only seven of these studies reported the R0 resection rate, and two of them were multi-arm studies. Interestingly, the results showed that neoadjuvant therapy

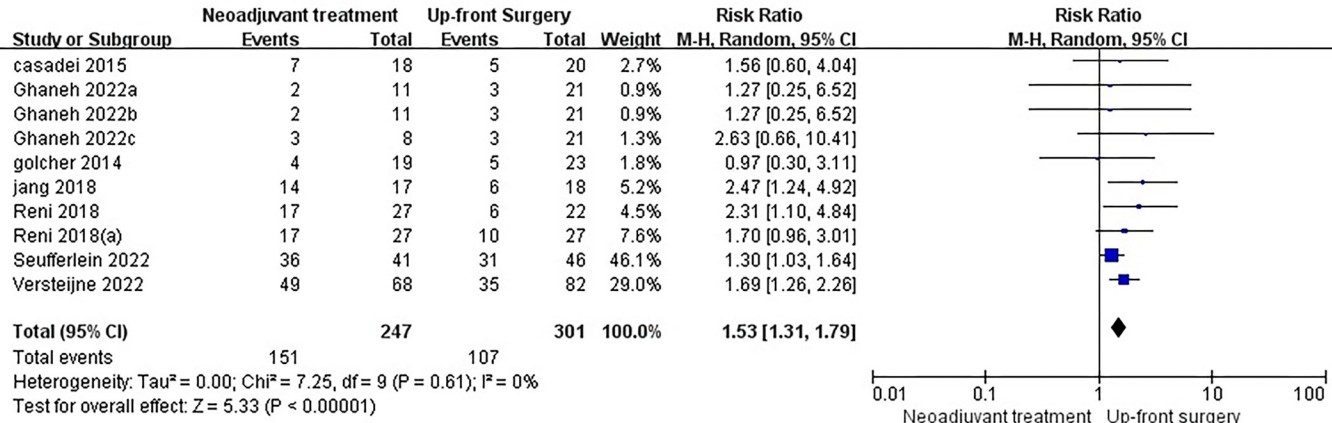

**Fig 5. Forest plots of R0 resection rate in patients with neoadjuvant therapy versus those surgery first.** It showed that neoadjuvant therapy significantly increased the R0 resection rate compared to surgery alone.

significantly increased the R0 resection rate compared to surgery alone (RR = 1.53, I2 = 0%, P< 0.00001) [Fig 5]. It is worth noting that the $I^2$ value was relatively low, indicating low heterogeneity among the included studies. Furthermore, the stability of the results was confirmed through sensitivity analyses, where the effect size and fixed model were modified. To further explore the efficacy of different neoadjuvant therapy regimens, a Bayesian network meta-analysis was conducted. The rankings of the nine competing neoadjuvant therapy regimens in terms of R0 resection were summarized. The analysis suggested that gemcitabine + cisplatin (Gem +Cis) + Radiotherapy had a higher likelihood of being the most favorable regimen [Fig 6]. This conclusion was drawn based on the fact that there were no significant differences observed among the individual studies, although the amount of data was limited [Table 2]. In comparison, Gem+Radiotherapy was ranked as the second best regimen in terms of R0 resection.

## Median overall survival

Data on the median overall survival (mOS) was obtained from a total of eight studies. In cases where the hazard ratio (HR) was not reported, we utilized a method outlined by Tierney and colleagues [17] to estimate them from summary statistics. Within the subgroup of studies that

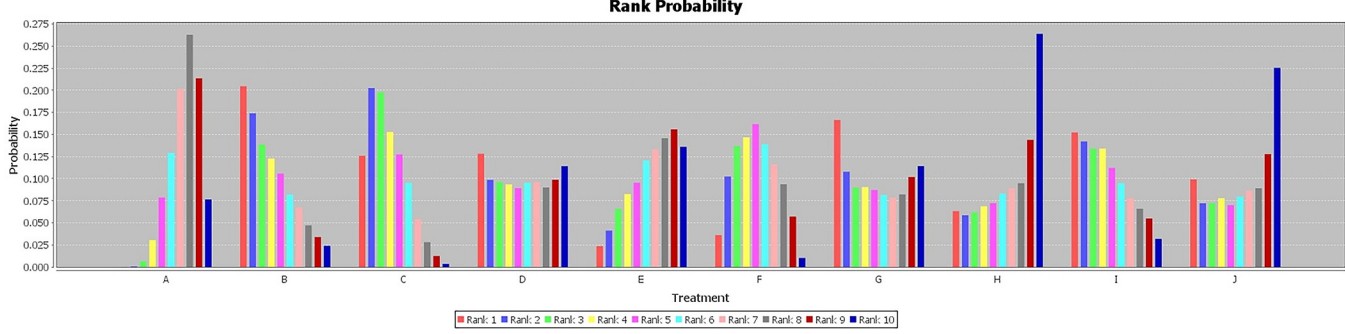

**Fig 6. The rank probability of R0 resection rate in patients with neoadjuvant therapy versus those surgery first.** The rankings of the nine competing neoadjuvant therapy regimens in terms of R0 resection were summarized. The analysis suggested that gemcitabine + cisplatin (Gem+Cis) + Radiotherapy had a higher likelihood of being the most favorable regimen. A: Surgery First; B: Gemcitabine + Cisplatin + Radiotherapy; C: Gemcitabine + Radiotherapy; D: Gem alone; E: Gemcitabine + Cisplatin then oral capecitabine; F: Gemcitabine + nab-paclitaxel; G: Ooxaliplatin + irinotecan + 5-fluorouracily; H: Gemcitabine + nab-paclitaxel + hydroxychloroquine; I: Oxaliplatin + leucovorin + irinotecan; J: gemcitabine + capecitabine.

**Table 2. The rank of R0 resection rate.**

| A | 3.72 (0.42, 36.38) | 3.66 (0.86, 18.02) | 1.99 (0.11, 33.93) | 1.21 (0.18, 8.97) | 2.25 (0.22, 21.16) | 2.20 (0.05, 80.10) | 1.05 (0.03, 36.87) | 3.06 (0.22, 31.54) | 1.30 (0.05, 31.33) |
|---|---|---|---|---|---|---|---|---|---|
| 0.27 (0.03, 2.40) | B | 1.00 (0.11, 8.92) | 0.54 (0.01, 19.34) | 0.32 (0.02, 6.15) | 0.60 (0.02, 13.70) | 0.59 (0.01, 38.11) | 0.28 (0.00, 18.95) | 0.81 (0.02, 20.12) | 0.35 (0.01, 16.29) |
| 0.27 (0.06, 1.17) | 1.00 (0.11, 8.87) | C | 0.54 (0.02, 12.21) | 0.33 (0.03, 3.77) | 0.61 (0.03, 8.36) | 0.59 (0.01, 27.07) | 0.28 (0.00, 12.85) | 0.81 (0.03, 12.18) | 0.34 (0.01, 11.32) |
| 0.50 (0.03, 8.91) | 1.86 (0.05, 75.68) | 1.84 (0.08, 50.56) | D | 0.60 (0.02, 19.69) | 1.13 (0.03, 42.74) | 1.11 (0.01, 111.30) | 0.53 (0.01, 48.67) | 1.53 (0.03, 64.72) | 0.65 (0.01, 45.97) |
| 0.83 (0.11, 5.66) | 3.08 (0.16, 60.79) | 3.06 (0.27, 38.99) | 1.66 (0.05, 48.02) | E | 1.85 (0.08, 35.96) | 1.82 (0.02, 112.46) | 0.87 (0.01, 50.63) | 2.52 (0.09, 52.03) | 1.06 (0.02, 43.55) |
| 0.44 (0.05, 4.53) | 1.66 (0.07, 43.61) | 1.65 (0.12, 29.27) | 0.89 (0.02, 33.75) | 0.54 (0.03, 12.03) | F | 0.99 (0.06, 16.07) | 0.45 (0.03, 7.76) | 1.33 (0.11, 12.91) | 0.57 (0.01, 28.80) |
| 0.46 (0.01, 18.56) | 1.71 (0.03, 132.45) | 1.69 (0.04, 104.21) | 0.90 (0.01, 104.62) | 0.55 (0.01, 41.54) | 1.01 (0.06, 16.95) | G | 0.46 (0.01, 25.52) | 1.35 (0.03, 57.19) | 0.58 (0.01, 78.91) |
| 0.95 (0.03, 35.13) | 3.54 (0.05, 290.48) | 3.58 (0.08, 203.96) | 1.89 (0.02, 199.04) | 1.15 (0.02, 76.50) | 2.21 (0.13, 36.49) | 2.15 (0.04, 110.22) | H | 2.94 (0.07, 104.01) | 1.21 (0.01, 145.88) |
| 0.33 (0.03, 4.54) | 1.23 (0.05, 44.06) | 1.23 (0.08, 29.59) | 0.65 (0.02, 30.91) | 0.40 (0.02, 11.09) | 0.75 (0.08, 8.73) | 0.74 (0.02, 30.41) | 0.34 (0.01, 14.51) | I | 0.43 (0.01, 30.50) |
| 0.77 (0.03, 19.98) | 2.83 (0.06, 156.28) | 2.91 (0.09, 100.75) | 1.53 (0.02, 116.88) | 0.94 (0.02, 42.30) | 1.77 (0.03, 79.35) | 1.73 (0.01, 183.90) | 0.83 (0.01, 89.11) | 2.34 (0.03, 114.20) | J |

A: Surgery First; B: Gemcitabine + Cisplatin + Radiotherapy; C: Gemcitabine + Radiotherapy; D: Gem alone; E: Gemcitabine + Cisplatin then oral capecitabine; F: Gemcitabine + nab-paclitaxel; G: Ooxaliplatin + irinotecan + 5-fluorouracily; H: Gemcitabine + nab-paclitaxel + hydroxychloroquine; I: Oxaliplatin + leucovorin + irinotecan; J: gemcitabine + capecitabine

specifically compared neoadjuvant therapy (NT) with upfront surgery (US), a total of four studies were included. The results of these studies indicated that neoadjuvant therapy led to a significant improvement in mOS when compared to upfront surgery (HR 0.68, 95% CI 0.58–0.92; P = 0.012; I2 = 15%) [Fig 7]. All neoadjuvant regimens were considered in the analysis, and no significant differences were observed among the various therapies. However, it appeared that the regimen including PEXG showed the most promising results, albeit with a relatively small sample size [Table 3].

## Major surgical complications

The incidence of significant surgical complications was reported in six studies, which consisted of two multi-arm studies. The rates ranged from 11% to 56% for NT and 11% to 45% for US. Interestingly, there was no statistically significant difference observed between NT and US, despite a considerable level of heterogeneity (RR = 0.96, 95%CI = 0.65–1.43; P = 0.85; $I^2$ = 46%) (Fig 8).

## Discussion

In our comprehensive network meta-analysis of randomized controlled trials (RCTs), we examined the effectiveness of neoadjuvant therapy (NT) compared to upfront surgery (US) for the treatment of pancreatic adenocarcinoma. Additionally, we explored the impact of different NT regimens when followed by surgery. Our findings revealed that NT showed significant improvements in overall survival (mOS) and the rate of complete tumor removal (R0 resection) compared to US in patients with pancreatic carcinoma. Among the various NT regimens, the combination of Gemcitabine, Cisplatin, and Radiotherapy (Gem+Cis+Radiotherapy) emerged as the most favorable in terms of achieving R0 resection. On the other hand, the

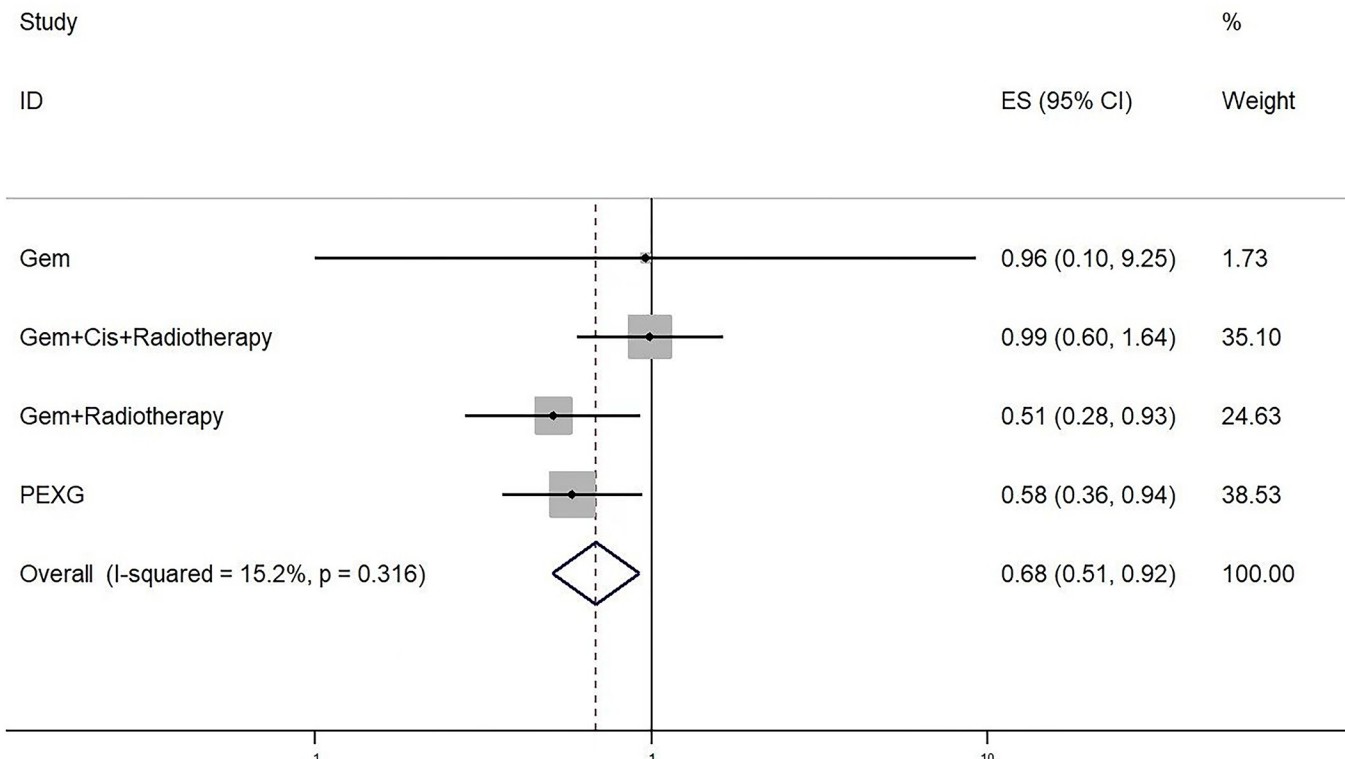

**Fig 7. Forest plots of median overall survival in patients with neoadjuvant therapy versus those surgery first.** Neoadjuvant therapy led to a significant improvement in mOS when compared to upfront surgery.

**Table 3. Rank of median overall survival.**

| Fixed Model | mean | sd | MC_error | val2.5pc | median | val97.5pc | start |
|---|---|---|---|---|---|---|---|
| hr[2] | 0.8504 | 0.04626 | 2.71E-04 | 0.7623 | 0.8494 | 0.9443 | 5000 |
| hr[3] | 1.109 | 0.5975 | 0.003454 | 0.3656 | 0.9762 | 2.627 | 5000 |
| hr[4] | 1.003 | 0.1123 | 6.26E-04 | 0.801 | 0.9969 | 1.242 | 5000 |
| hr[5] | 0.6746 | 0.08045 | 4.90E-04 | 0.5318 | 0.6694 | 0.8456 | 5000 |
| hr[6] | 0.8459 | 0.1193 | 0.002311 | 0.636 | 0.837 | 1.101 | 5000 |
| hr[7] | 1.103 | 0.2811 | 0.00314 | 0.6566 | 1.068 | 1.749 | 5000 |
| hr[8] | 0.8391 | 0.1171 | 0.002271 | 0.6316 | 0.8309 | 1.089 | 5000 |
| rk[1,1] | 0 | 0 | 5.77E-3 | 0 | 0 | 0 | 5000 |
| rk[2,1] | 0.0027 | 0.05189 | 3.01E-04 | 0 | 0 | 0 | 5000 |
| rk[3,1] | 0.2219 | 0.4155 | 0.002407 | 0 | 0 | 1 | 5000 |
| rk[4,1] | 0.0032 | 0.05648 | 3.36E-04 | 0 | 0 | 0 | 5000 |
| rk[5,1] | 0.6642 | 0.4723 | 0.00363 | 0 | 1 | 1 | 5000 |
| rk[6,1] | 0.0301 | 0.1709 | 0.001306 | 0 | 0 | 1 | 5000 |
| rk[7,1] | 0.0243 | 0.154 | 9.79E-04 | 0 | 0 | 0 | 5000 |
| rk[8,1] | 0.05366 | 0.2253 | 0.001983 | 0 | 0 | 1 | 5000 |

1: Surgery First; 2: Gemcitabine + Radiotherapy; 3: Gemcitabine alone; 4: Gemcitabine + Cisplatin + Radiotherapy; 5: Cisplatin/ Epirubicin then Gemcitabine/ Capecitabine (PEXG); 6: Ooxaliplatin + irinotecan + 5-fluorouracily; 7: Ooxaliplatin + irinotecan + 5-fluorouracily + Radiotherapy; 8: Gemcitabine + nab-paclitaxel

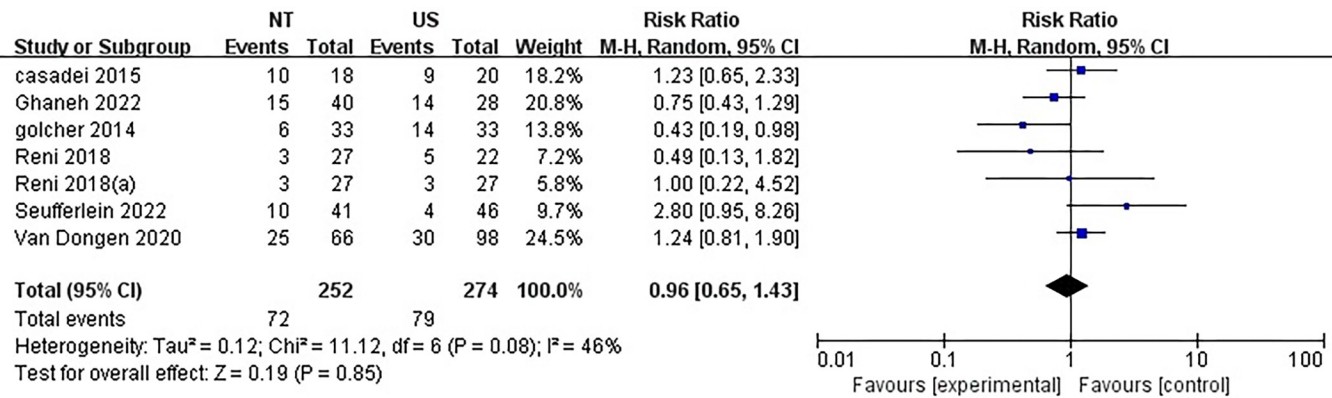

**Fig 8. Forest plots of complication in patients with neoadjuvant therapy versus those surgery first.** There was no statistically significant difference observed between NT and US, despite a considerable level of heterogeneity.

PEXG regimen demonstrated the highest likelihood of being ranked as the best in terms of mOS, despite having a relatively small sample size. It is worth noting that a previously published meta-analysis suggested no significant difference in overall survival [8], However, a recent study [22], which included important updates, provided evidence that NT can indeed improve overall survival for patients with pancreatic adenocarcinoma compared to upfront surgery. What sets our study apart is that we delved into identifying the optimal neoadjuvant therapy regimen, which significantly differed from previous research. Our analysis did not solely focus on comparing NT and US but rather encompassed a broader examination of various NT regimens. As a result, our study stands as the largest meta-analysis conducted in this specific context, providing valuable insights into the most effective treatment approach for pancreatic adenocarcinoma.

To date, there had been two published meta-analyses [33, 34] that had examined various neoadjuvant therapy (NT) regimens for pancreatic cancer. However, both studies had their limitations. The first study [33] did not include enough classical treatment regimens and had insufficient data analysis, the second study [34] had a relatively high heterogeneity due to the inclusion of unpublished studies and those only available as abstracts or conference presentations. Assessing the efficacy of neoadjuvant therapy for pancreatic cancer through randomized controlled trials (RCTs) had proven to be challenging [35, 36], leading to ongoing debates in the field. It was important to consider that different chemotherapy regimens might yield different therapeutic effects. In our analysis, we compared NT with upfront surgery (US) and examined the impact of different chemotherapy regimens on the likelihood of achieving R0 resection, with a particular focus on the Gem+Cis+Radiotherapy regimen. Interestingly, a study by Ghanem [22] also reached a similar conclusion, although they found that NT did not significantly increase the resection rate (RR 0.92). Furthermore, a recent abstract presentation at ASCO GI 2021 revealed that stereotactic body radiation therapy did not improve the R0 resection rate. However, NT was found to be significantly superior to neoadjuvant chemoradiotherapy in terms of overall survival (31 months vs. 17.1 months), disease-free survival (15 months vs. 10.2 months), and R0 resection rates (42% vs. 25%) [25]. These findings supported our analysis, which considered the impact of surgery first, chemoradiotherapy, and different chemotherapy regimens. Our results suggested that Gem+Cis+Radiotherapy might increase the likelihood of achieving R0 resection, indicating that radiotherapy played a role in improving surgical outcomes. In conclusion, our meta-analysis provides valuable insights into the efficacy of different neoadjuvant therapy regimens for pancreatic cancer. By considering various

factors, such as patient stage and tumor resectability, we have indirectly demonstrated the influence of radiotherapy on achieving R0 resection.

However, the majority of previous studies [22, 37] had consistently ranked neoadjuvant therapy (NT) higher than neoadjuvant chemoradiotherapy. A recent meta-analysis [34] also supported this finding, indicating that NT might be necessary for improving survival in pancreatic carcinoma. In contrast, our study found that neoadjuvant chemoradiotherapy ranked first, this was because neoadjuvant radiochemotherapy might benefit from certain local effects, such as achieving pN0 status. However, it was important to note that more clinical evidence was required to determine whether these local effects could translate into long-term outcomes. Additionally, investigating the potential benefits of additional radiotherapy in specific subgroups, such as those with hepatic artery (HA)/ superior mesenteric artery (SMA)/celiac axis (CA) contact between 90 and 180˚ or with N1/2 status, would be valuable.

In our study, the majority of the included studies focused on adjuvant therapy with various treatment regimens after surgery. However, we did not specifically analyze the impact of adjuvant therapy. It was possible that different adjuvant therapy regimens could have influenced the results. In the meantime, a recently published phase II clinical trial conducted on 147 patients randomized them to receive either perioperative mFOLFIRINOX or gemcitabinee-nab-paclitaxel. Unfortunately, this trial failed to improve upon the historical data from adjuvant therapy, as the 2-year overall survival rates were 47% and 48% for mFOLFIRINOX and gemcitabineenab-paclitaxel, respectively [28]. Another trial, the NEPAFOX trial, although inconclusive due to poor accrual, showed a potentially better median overall survival with adjuvant gemcitabine compared to neoadjuvant FOLFIRINOX for pancreatic carcinoma. Therefore, questions regarding the optimal neoadjuvant therapy regimen and the potential benefits compared to adjuvant (m)FOLFIRINOX remain unanswered. Therefore, Several ongoing randomized controlled trials, such as NorPACT-1, PANACHE01-PRODIGE48, PRE-OPANC-3, and Alliance A021806, are currently evaluating neoadjuvant therapy compared to upfront surgery followed by chemotherapy. The results of these trials will provide valuable insights and help resolve these lingering issues [38, 39].

The realm of major complications, it was found that the use of US NT did not lead to an increase in the rate of major complications. The results showed a medium level of heterogeneity (RR = 0.96, 95%CI = 0.65–1.43; P = 0.85; $I^2$ = 46%), indicating that result was stable. Interestingly, a study by Yamaguchi et al. [31] revealed that the FOLFIRINOX group had a higher relative dose intensity and a significantly lower rate of grade 3 or 4 adverse events compared to the GEM/nab-PTX group. This difference could be attributed to the preventive administration of, which was not given to patients receiving GEM/nab-PTX. This suggested that pegfilgrastim might have played a role in enhancing the feasibility and intensity of FOLFIRINOX treatment. Furthermore, golcher et al. [23] suggested that chemoradiation therapy might lead to less severe complications due to the induction of fibrosis, which improved the suitability of pancreatic tissue for anastomosis. A recent meta-analysis also found similar perioperative morbidity rates with and without NT [12]. In our meta-analysis, we found that NT did not have a significant impact on complications. However, it did prove to be a valuable tool for selecting patients for surgery. Preoperative therapy could help identify patients with initially unknown distant metastases, allowing them to be spared from unnecessary surgery [40].

## Limitations and advantages

One major limitation of our meta-analysis was the inclusion of studies with small sample sizes and short follow-up periods. Additionally, the use of different adjuvant chemotherapy regimens in the included studies was not taken into account, and we did not analyze the impact of

adjuvant chemotherapy after surgery. Furthermore, the majority of adjuvant therapy used gemcitabine monotherapy, which is now considered outdated. In the Netherlands, gemcitabine was the standard of care at the time of the trials. However, new evidence has emerged since then. For instance, the ESPAC-4 trial in 2017 demonstrated that adjuvant gemcitabine with capecitabine was more effective than gemcitabine monotherapy [41]. Similarly, the PRO-DIGE-24/CCTG PA.6 trial in 2018 showed that adjuvant fluorouracil, leucovorin, irinotecan, and oxaliplatin (FOLFIRINOX) was superior to adjuvant gemcitabine [42]. Despite these limitations, our meta-analysis had notable strengths, including a large number of patients, low heterogeneity in the results, and the inclusion of recently published clinical trials.

## Conclusion

In current comprehensive analysis of twelve randomized controlled trials (RCTs), it had been conclusively demonstrated that neoadjuvant therapy offered significant advantages for patients with resectable and borderline resectable pancreatic carcinoma. However, there was still some uncertainty surrounding the potential benefits of radiotherapy in improving the prognosis of pancreatic carcinoma. Therefore, it was imperative that future studies focus on comparing the efficacy of Gem+Cis+Radiotherapy with other neoadjuvant approaches, such as FOLFIRINOX or gemcitabine-based treatments. By doing so, we could gain a better understanding of which treatment strategy yields the best outcomes for patients.

## Supporting information

**S1 Data.**
(XLSX)

## Author Contributions

**Conceptualization:** Maoling Qin.

**Data curation:** Lu Huan, Fucai Yu.

**Formal analysis:** Hantao Zhou, Maoling Qin.

**Investigation:** Ding Cao.

**Methodology:** Yang Cao.

**Resources:** Lu Huan, Fucai Yu, Maoling Qin.

**Software:** Lu Huan, Hantao Zhou.

**Validation:** Yang Cao.

**Visualization:** Yang Cao.

**Writing – original draft:** Ding Cao, Yang Cao.

**Writing – review & editing:** Lu Huan, Yang Cao.

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
