## [Decision Letter · Decision Letter 0]

2 Oct 2023

PONE-D-23-23777Comparison of neoadjuvant treatment and surgery first for resectable or borderline resectable pancreatic carcinoma: a systematic review and network meta-analysis of randomized controlled trialsPLOS ONE

Dear Dr. Cao,

Thank you for submitting your manuscript to PLOS ONE. After careful consideration, we feel that it has merit but does not fully meet PLOS ONE’s publication criteria as it currently stands. Therefore, we invite you to submit a revised version of the manuscript that addresses the points raised during the review process.

We look forward to receiving your revised manuscript.

Kind regards,

Calogero Casà

Academic Editor

PLOS ONE

Journal Requirements:

5. Please ensure that you refer to Figure 4 in your text as, if accepted, production will need this reference to link the reader to the figure.

Reviewers' comments:

Reviewer's Responses to Questions

**Comments to the Author**

1. Is the manuscript technically sound, and do the data support the conclusions?

Reviewer #1: Yes

Reviewer #2: Yes

2. Has the statistical analysis been performed appropriately and rigorously? 

Reviewer #1: Yes

Reviewer #2: I Don't Know

3. Have the authors made all data underlying the findings in their manuscript fully available?

Reviewer #1: Yes

Reviewer #2: Yes

4. Is the manuscript presented in an intelligible fashion and written in standard English?

Reviewer #1: Yes

Reviewer #2: Yes

5. Review Comments to the Author

Reviewer #1: Given the right clarification on stereotactic radiotherapy in the "Discussion" (page 15, line 5), it could be made explicit that the chemoradiotherapy referred to conventional radiotherapy using standard dose.

In the "Conclusions", given the results of the systematic review conducted, in addition to the impact on the prognosis, it should be specified that the type of NT with the best impact on R0 after surgery is chemoradiotherapy, (otherwise, in my opinion, an important result could be eclipsed).

On page 9 line 12 "To" must be corrected to "o"

Reviewer #2: This sentence is generic and unclear: " Compared with surgery first, neoadjuvant therapy did increase the R0 resection rate (RR = 1.53, I2 = 0%, P＜ 0.00001), there was a certain possibility that gemcitabine + cisplatin (Gem+Cis) +Radiotherapy was the most favorable in terms of the fact that there was no significant difference concerning the results from the individual studies."

Also this sentence is unclear : "Furthermore, most published meta analyses reported higher R0 resection rates overall with neoadjuvant therapy, contradicting the clear benefit of neoadjuvant therapy"

As a radiation oncologist, I believe that the results are a little bit inconclusive: radiotherapy increases R0, this is a somewhat expected result, but does it improve survival? Do chemotherapy-only studies have better survival outcomes?

Eventually it would be worth underlining in the conclusions that, just as the studies using adjuvant gemcitabine as monotherapy are outdated, so too are the studies using conformal radiotherapy techniques, furthermore, the doses and times of radiotherapy can be an influencing factor.

Last comment: I am not an expert in statistics so tables 2 and 3 are a little difficult for me to understand, perhaps a better explanation would be needed

6. PLOS authors have the option to publish the peer review history of their article (what does this mean?). If published, this will include your full peer review and any attached files.

Reviewer #1: **Yes: **Elisabetta Lepre

Reviewer #2: No

---

## [Author Response · Author response to Decision Letter 0]

25 Oct 2023

Reviewer #1: Thank you for your nice comments on our article. According to your suggestions, we have make clear and corrected several mistakes in our previous draft. We have made extensive revisions to our previous draft. The detailed point-by-point responses are listed below. 

Comment 1: Given the right clarification on stereotactic radiotherapy in the "Discussion" (page 15, line 5), it could be made explicit that the chemoradiotherapy referred to conventional radiotherapy using standard dose.

Answer: Thanks for your suggestion. We have made a clear definition on stereotactic radiotherapy1 in our previous article and the correctional parts were highlighted by using red colored text. The correctional limitation was added in Page15 row 5-6.

 Comment 2: In the "Conclusions", given the results of the systematic review conducted, in addition to the impact on the prognosis, it should be specified that the type of NT with the best impact on R0 after surgery is chemoradiotherapy, (otherwise, in my opinion, an important result could be eclipsed).

Answer: Thanks for your suggestion. We have refined our conclusions better and the enhancive parts were highlighted by using red colored text. The enhancive limitation was added in Page19 row 10- 11.

 Comment 3: On page 9 line 12 "To" must be corrected to "o"

Answer: Thanks for your suggestion. We have corrected the mistake on page 9 line 12. 

Reviewer #2: Thank you for your valuable comments on our article. According to your suggestions, we have supplemented several parts to explain and corrected several mistakes in our previous draft. We have made extensive revisions to our previous draft. The detailed point-by-point responses are listed below.

Comment 1: This sentence is generic and unclear: " Compared with surgery first, neoadjuvant therapy did increase the R0 resection rate (RR = 1.53, I2 = 0%, P＜ 0.00001), there was a certain possibility that gemcitabine + cisplatin (Gem+Cis) +Radiotherapy was the most favorable in terms of the fact that there was no significant difference concerning the results from the individual studies." Also this sentence is unclear : "Furthermore, most published meta analyses reported higher R0 resection rates overall with neoadjuvant therapy, contradicting the clear benefit of neoadjuvant therapy"

Answer: Thanks for your suggestion. We make some optimizations to render these sentence more clear. 

Sentence from the first draft was changed to "Compared with surgery first, neoadjuvant therapy did increase the R0 resection rate (RR = 1.53, I2 = 0%, P＜ 0.00001), besides, by the net-work there was a certain possibility that gemcitabine + cisplatin (Gem+Cis) + Radiotherapy was the most favorable. " In Page 4 row 9- 10

Also, another sentence from the first draft was changed to "Furthermore, most published meta-analyses2-4 reported higher R0 resection rates but a lower surgical resection rate overall with neoadjuvant therapy, contradicting the clear benefit of neoadjuvant therapy." In Page 5 row 12- 13 and in Page 6 row 1

 Comment 2: As a radiation oncologist, I believe that the results are a little bit inconclusive: radiotherapy increases R0, this is a somewhat expected result, but does it improve survival? Do chemotherapy-only studies have better survival outcomes? Eventually it would be worth underlining in the conclusions that, just as the studies using adjuvant gemcitabine as monotherapy are outdated, so too are the studies using conformal radiotherapy techniques, furthermore, the doses and times of radiotherapy can be an influencing factor.

Answer: Thanks for your valuable suggestions. Based on studies those met our inclusion criteria, we considered that neoadjuvant chemoradiotherapy could increase R0, and by Bayesian network gemcitabine + cisplatin (Gem+Cis) + Radiotherapy had a higher likelihood of being the most favorable regimen. Similarly, we considered neoadjuvant chemoradiotherapy could improve mOS, but by Bayesian network Cisplatin/ Epirubicin then Gemcitabine/ Capecitabine (PEXG) might be relative best. Finally, as your valuable suggestions, outdated chemotherapy region and different radiation doses could make an influence on our results. We incorporated these deficiencies in our limitations and conclusion. In Page 19 row 1, 5- 6, 12- 13 and Page 20 row 1 

 Comment 3: I am not an expert in statistics so tables 2 and 3 are a little difficult for me to understand, perhaps a better explanation would be needed.

Answer: Thanks for your valuable suggestions. We have deleted some repetitive expression to make the data clearer and more concise. And we have re-uploaded tables 2 and 3, also a more detailed explanation have been added in tables.

---

## [Decision Letter · Decision Letter 1]

4 Dec 2023

Comparison of neoadjuvant treatment and surgery first for resectable or borderline resectable pancreatic carcinoma: a systematic review and network meta-analysis of randomized controlled trials

PONE-D-23-23777R1

Dear Dr. Cao,

We’re pleased to inform you that your manuscript has been judged scientifically suitable for publication and will be formally accepted for publication once it meets all outstanding technical requirements.

Kind regards,

Calogero Casà

Academic Editor

PLOS ONE

Additional Editor Comments (optional):

All comments have been addressed, the manuscript is of good quality, written in correct and fluent English. The data analysis is clear and leads transparently to the results, which are correctly interpreted in the discussion.

Reviewers' comments:

Reviewer's Responses to Questions

**Comments to the Author**

1. If the authors have adequately addressed your comments raised in a previous round of review and you feel that this manuscript is now acceptable for publication, you may indicate that here to bypass the “Comments to the Author” section, enter your conflict of interest statement in the “Confidential to Editor” section, and submit your "Accept" recommendation.

Reviewer #1: All comments have been addressed

Reviewer #2: All comments have been addressed

2. Is the manuscript technically sound, and do the data support the conclusions?

Reviewer #1: (No Response)

Reviewer #2: (No Response)

3. Has the statistical analysis been performed appropriately and rigorously? 

Reviewer #1: (No Response)

Reviewer #2: (No Response)

4. Have the authors made all data underlying the findings in their manuscript fully available?

Reviewer #1: (No Response)

Reviewer #2: (No Response)

5. Is the manuscript presented in an intelligible fashion and written in standard English?

Reviewer #1: (No Response)

Reviewer #2: (No Response)

6. Review Comments to the Author

Reviewer #1: (No Response)

Reviewer #2: (No Response)

7. PLOS authors have the option to publish the peer review history of their article (what does this mean?). If published, this will include your full peer review and any attached files.

Reviewer #1: No

Reviewer #2: No

---

## [Editor Report · Acceptance letter]

11 Dec 2023

PONE-D-23-23777R1 

Comparison of neoadjuvant treatment and surgery first for resectable or borderline resectable pancreatic carcinoma: a systematic review and network meta-analysis of randomized controlled trials 

Dear Dr. Cao:

I'm pleased to inform you that your manuscript has been deemed suitable for publication in PLOS ONE. Congratulations! Your manuscript is now with our production department. 

Kind regards, 

on behalf of

Dr. Calogero Casà 

Academic Editor

PLOS ONE